# The Association between Plasma Omega-6/Omega-3 Ratio and Anthropometric Traits Differs by Racial/Ethnic Groups and *NFKB1* Genotypes in Healthy Young Adults

**DOI:** 10.3390/jpm9010013

**Published:** 2019-02-16

**Authors:** Jeremy Bauman-Fortin, David W.L. Ma, David M. Mutch, Salma A. Abdelmagid, Alaa Badawi, Ahmed El-Sohemy, Bénédicte Fontaine-Bisson

**Affiliations:** 1School of Human Kinetics, Faculty of Health Sciences, University of Ottawa, Ottawa, ON K1N 6N5, Canada; jbaum073@uottawa.ca; 2Human Health & Nutritional Sciences, University of Guelph, Guelph, ON N1G 2W1, Canada; davidma@uoguelph.ca (D.W.L.M.); dmutch@uoguelph.ca (D.M.M.); salma.abdelmagid@gmail.com (S.A.A.); 3Public Health Risk Sciences Division, Public Health Agency of Canada, Toronto, ON M5V 3L7, Canada; alaa.badawi@canada.ca; 4Department of Nutritional Sciences, University of Toronto, Toronto, ON M5S 1A8, Canada; a.el.sohemy@utoronto.ca; 5School of Nutrition Sciences, University of Ottawa, Ottawa, ON K1N 6N5, Canada; 6Institut du savoir Montfort, Hôpital Montfort, Ottawa, ON K1K 0T2, Canada

**Keywords:** *NFKB1*, single nucleotide polymorphism, omega-6/omega-3 ratio, body mass index, waist circumference, racial/ethnic groups

## Abstract

Evidence for a relationship between omega-6/omega-3 (n-6/n-3) polyunsaturated fatty acid (PUFA) ratio and obesity in humans is inconsistent, perhaps due to differences in dietary intake or metabolism of PUFAs between different subsets of the population. Since chronic inflammation is central to obesity and inflammatory pathways are regulated by PUFAs, the objective of this study was to examine whether variants in the *NFKB1* gene, an upstream regulator of the inflammatory response, modify the association between the n-6/n-3 ratio (from diet and plasma) and anthropometric traits in a multiethnic/multiracial population of young adults. Participants’ (*n* = 898) dietary PUFA intake was assessed using a food frequency questionnaire and plasma PUFA concentrations by gas chromatography. Nine tag single nucleotide polymorphisms (SNP) in *NFKB1* were genotyped. Significant interactions were found between racial/ethnic groups and plasma n-6/n-3 ratio for body mass index (BMI) (*p* = 0.02) and waist circumference (WC) (*p* = 0.007). Significant interactions were also observed between racial/ethnic groups and three *NFKB1* genotypes (rs11722146, rs1609798, and rs230511) for BMI and WC (all *p* ≤ 0.04). Significant interactions were found between two *NFKB1* genotypes and plasma n-6/n-3 ratio for BMI and WC (rs4648090 *p* = 0.02 and 0.03; rs4648022 *p* = 0.06 and 0.04, respectively). Our findings suggest that anthropometric traits may be influenced by a unique combination of n-6/n-3 ratio, racial/ethnic background, and *NFKB1* genotypes.

## 1. Introduction

Essential polyunsaturated fatty acids (PUFAs) are vital to many human physiological processes such as membrane structure and function, signaling pathways, and gene expression [1]. Despite their vital role in the body, essential fats such as linoleic acid (LA), an omega-6 (n-6), and α-linolenic acid (ALA), an omega-3 (n-3), cannot be synthesized by the body and must be provided by the diet [1]. Both LA and ALA can be elongated and desaturated by the body into longer chain PUFAs, such as arachidonic acid (n-6), and eicosapentaenoic acid (EPA) and docosahexaenoic acid (DHA) (both n-3) [1]. Although the same desaturation enzymes regulate these conversions, the n-6 pathway (e.g., LA converted into arachidonic acid) is generally favored over the n-3 pathway (e.g., ALA into EPA and DHA) primarily due to higher n-6 levels in the body [1,2]. While humans have evolved eating an n-6/n-3 ratio closer to 1:1, the development and use of food technology in agriculture and the reduction of wild fish, animal, and plant consumption have resulted in an increase in the n-6/n-3 ratio ranging from 4:1 to 50:1, with an average of 20:1 [2,3]. The shift towards a higher n-6/n-3 ratio has been proposed to contribute to obesity development [2]. 

N-6 and n-3 fatty acids induce opposing effects on obesity-related traits including adipocyte maturation and growth, fat storage and oxidation, macrophage activation, chronic inflammation, brain–gut–adipose tissue signaling pathways, as well as hypothalamic energy homeostasis [2,4,5]. For example, several reviews of epidemiological studies showed that the relationship between higher intake of long-chain n-3 PUFAs in adults and the proportion of overweight/obese individuals is controversial, but n-6 erythrocyte concentrations or dietary intake is positively associated with prospective weight gain [2,4]. The review of clinical trials revealed that long-chain n-3 PUFAs increase energy expenditure, fat oxidation, and appetite suppression, but have little to no difference on weight loss [2,5,6,7]. In fact, a meta-analysis of randomized controlled trials investigating the effect of n-3 supplementation on obesity management showed an overall small decrease in waist circumference (WC), but no effect on weight loss and body mass index (BMI) [8]. Overall, findings from the relationship between n-6/n-3 ratio and obesity in humans are limited and inconsistent [4]. In mice, an n-6 (linoleic acid)-enriched diet resulted in increased adipogenesis and body weight gain when compared to mice fed a standard diet or an n-3 (ALA)-enriched diet [9]. Conversely, transgenic *Fat-1* mice, which are able to endogenously convert n-6 into n-3 PUFAs, and rodents fed an enriched n-3 diet were more resistant to weight gain, had decreased fat deposition, and higher energy expenditure [10,11,12]. Transgenic *Fat-1* mice were also shown to have reduced inflammation through a marked attenuation of the nuclear factor NF-κB activity [11], a transcription factor acting as an upstream regulator of the inflammatory response [13]. Lines of evidence suggest that one of the proposed mechanisms linking n-6 and n-3 PUFAs to obesity and obesity-related complications is inflammation [14]. Since NF-κB can be modulated by n-6 and n-3 fatty acids [15,16,17,18,19,20], variations in this gene may potentiate or attenuate the n-6/n-3 mediated inflammatory response and its subsequent effects on obesity-related mechanisms. 

The *NFKB1* gene encodes for the p105 and p50 of the NF-κB subunit which translocates from the cytoplasm to the nucleus to activate gene expression [21]. The functional variant -94ins/delATTG in the *NFKB1* gene was found to be more frequent (ins/ins genotype) in morbidly obese women compared to lean controls and was associated with higher C-reactive protein concentrations, an acute-phase inflammatory protein, than in morbidly obese carriers of the del genotype [22]. The risk of acute coronary syndrome was also highest among obese male carriers of the del allele of the same *NFKB1* -94ins/delATTG variant compared to nonobese and/or noncarriers of the variant allele [23]. Thus, we hypothesize that inconsistent results in human observational studies examining the relationship between PUFA intake and obesity-related phenotypes may partly be explained by genetic variations in *NF-κB* given its upstream role in inflammation and its known regulation by PUFAs. Thus, the goal of this study was to examine whether variants in the *NFKB1* gene modify the association between the n-6/n-3 ratio, both from dietary intake and plasma concentrations, and anthropometric traits (BMI and WC) in young adults. Identifying factors influencing anthropometric traits early on in adulthood might help to predict risk of obesity development and inform obesity prevention strategies.

## 2. Materials and Methods

### 2.1. Participants

Participants were young adults (*n* = 898; 638 women and 260 men) aged 20–29 years who took part in the Toronto Nutrigenomics and Health study. They were recruited from the University of Toronto campus between October 2004 and July 2009. Participants were asked to fill out a general health and lifestyle questionnaire (e.g., age, sex, smoking habits) which was also used to document their self-reported racial/ethnic group. This information was used to categorize them into three main groups: Caucasians (*n* = 455; Europeans, Middle Easterns, and Hispanics), East Asians (*n* = 338; Chinese, Japanese, Koreans, Filipinos, Vietnamese, Thais, and Cambodians), and South Asians (n = 105; Bangladeshis, Indians, Pakistanis, and Sri Lankans). Participants belonging to other racial/ethnic groups or more than 2 groups were categorized as “others” and not included in the study due to a large within-group diversity and variability in the main variables of interest (diet, genetics, and body composition). Physical activity was assessed using a questionnaire asking to report the types of modifiable physical activities (leisure or occupational activity) that they usually took part of on weekdays and weekend days. Participants had to specify the intensity and the number of hours spent for each activity, which was later converted into metabolic equivalent tasks (METs). Accordingly, light activities are equivalent to 2.4 METs, moderate activities to 3.5 METs, and vigorous activities to 7.5 MET. The daily score was then weighted for each weekday and weekend day and then averaged to yield a physical activity score in MET-hours per week. Participants were excluded from the study if they were pregnant or breastfeeding due to hormonal and physiological adaptations occurring during these periods [24]; if they had chronic diseases that could influence inflammation (e.g., type 1 or 2 diabetes, cancer, cardiovascular disease); or if they could not provide a venous blood sample. 

### 2.2. Anthropometric Traits

Body weight was measured to the nearest 0.1 kg using an electronic scale. A wall-mounted stadiometer was used to measure height to the nearest 0.1 cm. Body mass index (BMI) was calculated using the following equation: weight (kg)/(height (m))^2^. Waist circumference (WC) was measured using a nonstretchable measuring tape at the midpoint between the lower ribs and the iliac crest. At least two measurements were taken and, if they differed by more than 1 cm, a third one was taken. The mean of WC measurements was used for the analyses.

### 2.3. Dietary Assessment

Participants completed a Toronto-modified, 196-item Willett semiquantitative food frequency questionnaire (FFQ), reporting their dietary intake over the past month. Thorough instructions as well as visual aid of portion sizes were provided to all participants to improve the assessment. Declared intake for each food item was then converted to daily number of servings and used to estimate total dietary energy intake, as well as nutrient intake including n-6 and n-3 fatty acids. Individuals who declared an unlikely energy intake of <800 kcal/day or >3500 kcal/day for women and >4000 kcal/day for men were categorized as potential under- and over-reporters and were excluded (*n* = 80) from the analyses as previously described [25]. One unrealistic dietary intake value of n-6 fatty acid (780 g; 124 times the standard error) was deleted from the dataset. 

### 2.4. Measurement of Plasma n-6 and n-3 Concentrations

Following a minimum fast of 12 hours, blood samples were collected from the antecubital vein at LifeLabs medical laboratory services (Toronto, Canada). Blood samples were centrifuged to isolate plasma, which was subsequently frozen at −80 °C until analysis. After being thawed, gas chromatography was used to quantify total plasma fatty acid concentrations, as previously described [26]. C17:0 was used as an internal standard to calculate fatty acid concentrations [26]. For our analyses, n-3 PUFAs included alpha-linolenic acid, stearidonic acid, eicosatrienoic acid, eicosapentaenoic acid, docosapentaenoic acid, and docosahexaenoic acid; and n-6 PUFAs included linoleic acid, γ-linoleic acid, eicosadienoic acid, di-homo-γ-linoleic acid, arachidonic acid, docosadienoic acid, adrenic acid, and osbond acid.

### 2.5. Single Nucleotide Polymorphism Selection and Genotyping

DNA was isolated from white blood cells and multiplex genotyping was performed using the Sequenom MassArray^®^ platform at Princess Margaret Hospital. Tag-single nucleotide polymorphism (SNP) selection in the *NFKB1* gene was performed using HapMap release 27 (https://www.genome.gov/10001688/international-hapmap-project/) in the population with European ancestry (CEU) and Haploview 4.2 (https://www.broadinstitute.org/haploview/haploview) with a minimum minor allele frequency of 5% and *r^2^* threshold of 0.80 using pairwise tagging. Eighteen tag SNPs in the *NFKB1* gene were identified and genotyped in this cohort. For quality control, 10% of the population was genotyped a second time and a >99% concordance was achieved. Since we had a multiethnic population and our tag SNPs were chosen based on a population with European ancestry in HapMap, we tested the linkage disequilibrium (LD) in our entire sample and within each racial/ethnic group. LD plots were generated using the Linkage Format in Haploview 4.2, with an *r^2^* threshold of 0.80. To detect any potential genotyping errors or population stratification, deviation from Hardy–Weinberg Equilibrium (HWE) using a *χ*-square test with 1 degree of freedom was tested in the entire sample and by racial/ethnic group. 

Two SNPs deviated from HWE in all participants (SNPs rs3774934 and rs3774956); however, they were in HWE within individual racial/ethnic groups. Rs4698863 deviated from HWE in all participants as well as within the South Asian group. Since these three SNPs were in high LD (≥95%) with SNPs rs3774932, rs1599961, and rs767464, respectively, they were removed from the analyses. Other SNPs that were in high LD (≥95%) with at least one other SNP in all participants as well as in all racial/ethnic groups were also removed (rs13117745, rs1609798, rs230547, rs3774934, rs3774956, rs4648110, and rs4648217). We therefore genotyped 18 SNPs but used 9 for the current analysis (rs11722146, rs13117745, rs1609798, rs4648022, rs4648090, rs1599961, rs230511, rs7674640, rs3774932).

### 2.6. Statistical Analyses

The Statistical Analysis System© (SAS version 9.4) software was used to conduct all statistical analyses. Any participants with missing information for the predictors, outcomes, and covariates of interest were excluded from the analyses (original sample *n* = 1635, plasma PUFA measured in *n* = 1088, final sample size of *n* = 898). Descriptive statistics, including analysis of variable distribution and potential outliers, were done on dependent and independent variables. Confounder analysis was performed by examining the relationship between age, sex, physical activity, ethnicity, smoking, caloric intake, and C-reactive protein measurements (sign of inflammation), and the independent (n-6/n-3 ratio and genotypes) as well as the dependent variables (BMI and WC). Variables associated with one of the predictors (n-6/n-3 or SNPs) and one of the outcomes (BMI or WC) were entered in the statistical model, that is, sex, age, physical activity level, caloric intake, and ethnicity. 

Since the distribution of values was skewed for the dietary and plasma n-6/n-3 ratios, these variables were divided into two groups: lower and higher dietary and plasma concentrations of n-6/n-3 ratio groups using the median (8.66 and 12.97, respectively). General characteristics of the participants were then compared between the plasma lower vs higher n-6/n-3 groups. Continuous and normally distributed variables were compared using a Student *t*-test (mean ± standard error (SE)), continuous but non-normally distributed variables were compared using a Wilcoxon rank sum test or Kruskal–Wallis test (median ± interquartile range), and categorical variables using a chi-square or Fisher’s exact test. All values are reported as mean ± SE unless otherwise stated. Since the chi-square test significantly differed between the plasma n-6/n-3 ratio and racial/ethnic groups, pairwise comparisons were performed using the Bonferroni correction (all *p* were <0.0001). Spearman correlations assessed the correlations between dietary and plasma concentrations of PUFAs. General linear models (GLMs) were used to assess the associations between our predictor (dietary and plasma n-6/n-3 ratios, and SNPs) and outcome variables (BMI and WC), as well as gene–diet interactions, adjusted for calories, physical activity, age, sex, and racial/ethnic groups. Some interactions between race/ethnicity and the predictors (n-6/n-3 or SNPs) were observed, so these analyses were instead stratified by racial/ethnic groups. As expected, BMI and WC differed by sex. However, since no interaction between sex and other predictors of interest (n-6/n-3, SNPs) nor a three-way interaction (sex, n-6/n-3, SNPs) were found, both men and women were pooled and adjusted for in all analyses. SNPs were analyzed in an additive model except for rs4648022, which had a minor allele frequency (MAF) of 5%. When analyses were further stratified by racial/ethnic groups, the rs13117745 (MAF of 12%) was analyzed as a dominant model since some groups contained five or less individuals. The rs4648090 SNP (MAF of 7%) and rs4648022 were not analyzed when stratified by *NFKB1* genotype and race/ethnicity groups since the dominant model still had less than five individuals in some groups. Multiple comparisons between groups within each statistical model were accounted for using a Tukey post hoc test. The overall issue of multiple hypothesis testing was dealt with using the False Discovery Rate. Significant differences were set at a *p* value <0.05.

## 3. Results

### 3.1. General Characteristics

The majority of the sample consisted of Caucasian women in their early twenties who were nonsmokers and lightly active (Table 1). Most of the participants fell within the least-risk category for BMI (i.e., 18.5–24.9 kg/m^2^) and WC (i.e., <102 cm for men and <88 cm for women) according to Canadian guidelines for body weight classifications in adults [27]. When comparing the lower with the higher plasma concentrations of n-6/n-3 ratio groups (based on the median), none of the general characteristics differed except for the distribution of racial/ethnic groups (*p* < 0.0001; Table 1). All three racial/ethnic group frequencies significantly differed from each other (increased frequency of higher n-6/n-3 ratio Caucasians ˃ East Asians ˃ South Asians, all *p* < 0.0001 for pairwise comparisons; Table 1).

Among our young and generally healthy participants, the mean (9.4) and median (8.7) of dietary n-6/n-3 ratio were lower than the average reported ratio of 20:1 for Western dieters [3]. As expected, dietary n-6 and n-3 intake and their ratio significantly differed between the plasma lower vs higher n-6/n-3 ratio groups (all *p* ≤ 0.04, Table 1). However, while plasma n-6 concentrations did not differ between groups (*p* = 0.22), significantly lower n-3 concentrations were observed in the higher n-6/n-3 ratio group (*p* = 0.0001), thus explaining the significant difference in the overall plasma n-6/n-3 ratio (*p* < 0.0001; Table 1). The plasma concentrations of the n-6/n-3 ratio also differed between racial/ethnic groups (median and interquartile range (IQR) for Caucasians 13.5 ± 4.6, East Asians 11.8 ± 4.6, and South Asians 13.1 ± 4.4; *p* < 0.0001), and similarly for dietary n-6/n-3 ratio (Caucasians 8.9 ± 3.0, East Asians 8.5 ± 3.0, and South Asians 8.6 ± 3.4; *p* = 0.01). Finally, correlations were found between dietary intakes of n-6/n-3 ratio and plasma concentrations of n-6/n-3 ratio (*r* = 0.34, *p* < 0.0001) and marginally for n-3 (*r* = 0.07, *p* = 0.05) but not for n-6 (*r* = −0.02; *p* = 0.57) PUFAs. 

### 3.2. Associations between n-6/n-3 Ratios, NFKB1 Genotypes, and Anthropometric Traits

No difference in BMI and WC was observed between plasma lower and higher n-6/n3 ratio groups (*p* = 0.76 and 0.82, respectively) nor for the dietary lower and higher n-6/n-3 ratio groups (*p* = 0.66 and 0.16, respectively) when adjusted for age, sex, caloric intake, and physical activity. However, significant interactions were observed between plasma lower and higher n-6/n-3 ratio and racial/ethnic groups for BMI (*p* = 0.02) and for WC (*p* = 0.007). No interaction was found for dietary n-6/n-3 ratio (*p* = 0.95 for BMI and *p* = 0.99 for WC). Among the lower plasma n-6/n-3 ratio groups, Caucasians had significantly higher mean BMI and WC than East Asians (23.9 ± 0.2 vs. 22.1 ± 0.2 kg/m^2^; and 78.2 ± 0.6 vs. 73.3 ± 0.5 cm, respectively, both *p* < 0.0001, Figure 1). Among the higher n-6/n-3 ratio group, East Asians had lower BMI and WC (21.8 ± 0.3 kg/m^2^ and 72.1 ± 0.6 cm, respectively) than Caucasians (23.2 ± 0.2 kg/m^2^ and 76.8 ± 0.5 cm, respectively, both *p* ≤ 0.0006) and South Asians (24.3 ± 0.4 kg/m^2^ and 78.5 ± 1.0 cm, respectively, both *p* < 0.0001, Figure 1). Consistent with the continuous dietary intakes and plasma concentrations of the n-6/n-3 ratio differing between racial/ethnic groups, the distribution also differed between the lower vs higher n-6/n-3 ratio groups (Table 1). Significant differences between racial/ethnic groups were also found for BMI and WC when adjusted for caloric intake, physical activity, age, and sex (both *p* < 0.0001). Both Caucasians (23.5 ± 0.2 kg/m^2^ and 77.4 ± 0.4 cm) and South Asians (23.7 ± 0.3 kg/m^2^ and 76.8 ± 0.7 cm) had comparable BMI and WC, respectively, which were both significantly higher than that of East Asians (22.0 ± 0.2 and 72.7 ± 0.4 cm, both *p* < 0.0001). 

A description of the polymorphisms in the *NFKB1* gene (reference numbers and alleles, positions), as well as minor allele and genotype frequencies are shown in Appendix A. None of the genotypes were associated with differences in BMI or WC. However, *NFKB1* genotype frequency significantly differed by racial/ethnic groups for rs11722146, rs1609798, rs230511, rs4648022, rs4648090 and rs13117745, rs1599961 (Appendix A). Additionally, significant interactions were found between three *NFKB1* genotypes (rs11722146, rs1609798, and rs230511) and racial/ethnic groups for BMI (*p* = 0.02, 0.003, and 0.01, respectively) and WC (*p* = 0.04, 0.02, and 0.03, respectively), thus analyses were further stratified by racial/ethnic groups (Table 2). When examined within racial/ethnic groups, significant differences in BMI (all *p* ≤ 0.02) with similar trends for WC (all *p* ≤ 0.09) were observed between genotypes within South Asians but not among Caucasians and East Asians (Table 2).

### 3.3. Interactions between Lower and Higher n-6/n-3 Ratio Groups and NFKB1 Genotypes on Anthropometric Traits

Since no three-way interactions were found between the *NFKB1* genotypes, lower and higher n-6/n-3 ratio, and racial/ethnic groups, analyses were performed among all participants and adjusted for ethnicity in addition to the other covariates. Significant interactions were found between the *NFKB1* rs4648090 genotypes and plasma n-6/n-3 ratio groups on BMI and WC (*p* = 0.02 and 0.03, respectively) and similarly for rs4648022 (*p* = 0.06 and 0.04, respectively; Table 3). Patterns of anthropometric traits by n-6/n-3 ratio groups and genotypes on anthropometric traits are shown in Figure 2 (Panel A for rs4648090, Panel B for rs4648022). No interactions were found with dietary n-6/n-3 ratio groups and SNPs on BMI and WC. None of all the above-mentioned results remained significant after applying the false discovery rate. 

## 4. Discussion

This study aimed to examine the effect of variants in the *NFKB1* gene on the association between the n-6/n-3 ratio, both from dietary intake and plasma concentrations, and anthropometric traits (BMI and WC) in multiracial/ethnic population of young adults. We found significant differences in dietary and plasma n-6/n-3 ratio, *NFKB1* genotype frequencies, and anthropometric traits between racial/ethnic groups. East Asians were more likely to have lower BMI and WC as well as lower n-6/n-3 ratio compared to Caucasians and South Asians. Significant differences in BMI were also observed between *NFKB1* genotypes for three variants (rs11722146, rs1609798, and rs230511) but only among South Asians. Finally, significant interactions were found between the lower vs higher plasma n-6/n-3 ratio groups and two *NFKB1* genotypes (rs4648090 and rs4648022) on BMI and WC. None of the interactions were significant with dietary n-6/n-3 ratio and results were no longer significant when adjusted for multiple testing. 

Animal and human studies suggest that an increased n-6/n-3 ratio may play a role in the development of obesity through various pathways including adipogenesis, insulin and leptin resistance, activation of the cannabinoid system and, more relevant to our candidate gene of interest, chronic inflammation [3]. While we did not identify an association between n-6/n-3 ratio and anthropometric traits in our population of young adults, we did observe differences in the relationship between racial/ethnic groups. Differences in intakes of PUFAs, as well as anthropometric traits, between racial/ethnic groups have been previously documented. For example, in the large American MESA multiracial/ethnic prospective cohort of older US adults (*n* = 2837), individuals in the highest quartile of plasma seafood-derived n-3 fatty acids were more likely to be Chinese Americans, have a lower BMI, and have higher fruit and vegetables but lower meat intakes; while those in the highest quartile of plasma n-6 were younger, less physically active, and had lower consumption of whole grains, fish, fruit, and vegetables [28]. In this same cohort, plasma long-chain n-3 (EPA and DHA) concentrations were inversely associated with markers of inflammation and with a lower risk of cardiovascular disease. However, n-3 docosapentaenoic acid concentrations (DPA largely derived from endogenous metabolism) were inversely associated with cardiovascular risk only among Caucasians and Chinese Americans [28]. Using the National Health and Nutrition Examination Survey (NHANES) data, Heymsfield et al. showed that for individuals with the same BMI and height, regional mass and body composition (e.g., fat, musculoskeletal and lean mass) differed between races/ethnicities, and the difference was stronger among young individuals [29]. Indeed, lower cut-off points for higher-risk categories of BMI and WC have been proposed for Asian populations [30]. In contrast to Western diets, diets in India are mostly vegetarian and very low in fat intake (including low n-3 and negligible long-chain n-3 intake) and high in n-6 fatty acids [31]. An unbalanced n-6/n-3 ratio, among many other factors, may have played a role in the dramatic shift in the prevalence of obesity (˃24-fold) in India in less than half a century [31]. Nevertheless, trying to identify which factors are responsible for differences in anthropometric traits between racial/ethnic groups is extremely challenging, since body composition and stature are influenced by a myriad of interrelated factors including genetics, early life morphogenesis and aging, diet, lifestyle, and environment [30]. 

Gene–diet interactions involving PUFAs have been suggested to contribute to disease in a population-specific manner since the frequency of polymorphisms affecting PUFA biosynthesis and metabolism, as well as dietary intake, differ between populations [32]. Obesity is characterized by a low-state chronic inflammation, which is itself influenced by environmental, lifestyle, behavioral, and genetic determinants [33]. Our findings are illustrative of these complex interactions. NF-κB is a key transcription factor that plays a central role in inducing the expression of hundreds of proinflammatory genes and can be activated or inhibited by n-6 and n-3 fatty acids, respectively [15,16,17,18,19,20]. We found interactions between three *NFKB1* variants (rs11722146, rs1609798, and rs230511) and BMI and similarly with WC, but the effect was only significant among South Asians. In our cohort, not only BMI significantly differed between racial/ethnic groups but also genotype frequencies for these SNPs. The rs1609798 variant has been associated with apolipoprotein B concentrations, a marker of cardiovascular disease [34]. The two other variants (s11722146 and rs1609798) have been associated with risk of cancer [35,36,37]. Finally, we found gene–diet interactions between n-6/n-3 ratio and two *NFKB1* genotypes (rs4648090 and rs4648022) on BMI and WC. In another study, the rs4648022 polymorphism, but not rs230511 nor rs11722146, was associated with decreased risk of non-Hodgkin’s lymphoma cancer [38]. The rs4648090 polymorphism has also been related to decreased risk of breast cancer among women with European ancestry [39] and to interact with other SNPs in the *IFNG, IRF2*, *IL6*, and *NFKB1* genes to affect risk of colon cancer, and with *IRF6* and *NFKB1* genes for risk of rectal cancer [36,40]. These findings emphasize that risk of disease is explained by a complex interplay between factors which are difficult to tease apart. 

There are several strengths and limitations to our study. The use of a multiracial/ethnic cohort enables us to verify our hypothesis in individuals with differing genetic, metabolic, and lifestyle backgrounds. The measurement of plasma PUFA concentrations is more representative of what tissues were exposed to and takes into account differences in absorption and synthesis of endogenous fats which cannot be estimated by FFQ. In a large multiethnic American cohort, de Oliverira Otto et al. used an FFQ and showed that dietary estimates of seafood-derived PUFAs were moderately (*r* of 0.41 to 0.46), but plant-derived PUFAs very weakly (*r* of 0.05 to 0.13), associated with plasma fatty acid concentrations [29]. We found a correlation between dietary and plasma n-6/n-3 ratio and marginally for n-3 but not for n-6 PUFAs. Given the nature of this cross-sectional study, we cannot confer causality between our predictors and outcomes of interest. In addition, BMI and WC are proxy measures of body composition and do not always reflect adiposity or risk of disease. We did not find an association between n-6/n-3 ratio and obesity-related anthropometric traits. However, our ability to assess this association may have been hampered by the limited distribution of BMI and WC in our population of healthy young individuals. Race/ethnicity was self-reported and grouped in broad categories (i.e., Caucasians, East Asians, South Asians). However, there will inevitably remain some within-group variability in their genetics, lifestyle (including diet), and body composition. In order to decrease variability in our sample, we excluded those with a mixed racial/ethnic background or belonging to smaller/other groups that could not be combined with our three main ones. Since the study sample was stratified in subgroups (PUFA ratio, racial/ethnic groups, genotypes), there was a loss of power that may have diluted true associations. Limited data exists on young and healthy individuals with normal anthropometric traits. From a prevention perspective, small differences in certain biomarkers or anthropometric traits might help predict the development of obesity or risk of chronic diseases.

## 5. Conclusions

Overall, our study suggests that anthropometric traits may be influenced by n-6/n-3 ratio but the association appears to differ by racial/ethnic groups and weakly by *NFKB1* genotypes. These findings may contribute to providing a better understanding of the multifactorial causes of interindividual variability in anthropometric traits which are partly influenced by genetics (including inflammatory processes) and lifestyle factors. 

## Figures and Tables

**Figure 1 jpm-09-00013-f001:**
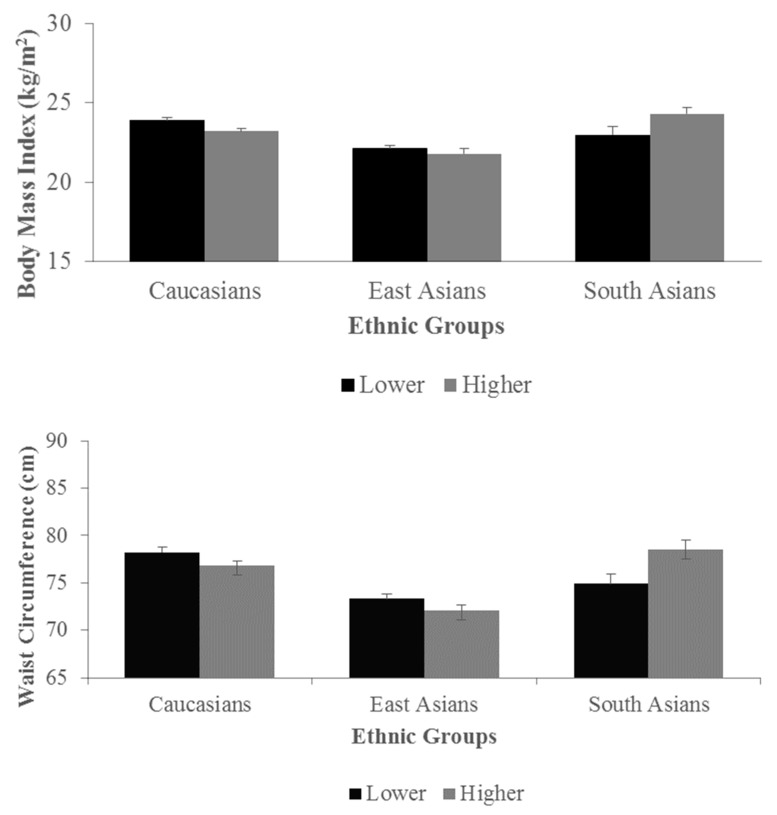
Interactions between the plasma lower (black—below the median) and higher (grey—above the median) n-6/n-3 ratio groups and racial/ethnic groups on BMI (*p* = 0.02) and WC (*p* = 0.007). *P* for interactions for differences in BMI or WC between racial/ethnic groups were tested using a general linear model adjusted for caloric intake, physical activity, age, and sex. Abbreviations: BMI, body mass index; WC, waist circumference.

**Figure 2 jpm-09-00013-f002:**
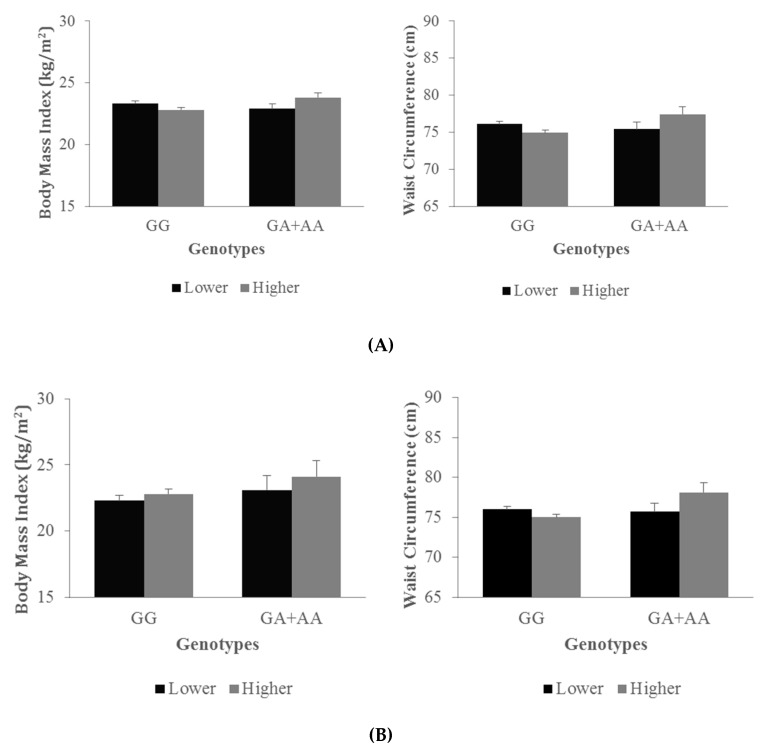
Interactions between the plasma lower (black—below the median) and higher (grey—above the median) n-6/n-3 ratio groups and *NFKB1* rs4648090 (**A**) and rs4648022 (**B**) genotypes on BMI (*p* = 0.02 and 0.06, respectively) and WC (*p* = 0.03 and 0.04, respectively). *P* for gene–diet interactions on BMI and WC were tested using a general linear model adjusted for caloric intake, physical activity, age, sex, and race/ethnicity. Abbreviations: BMI, body mass index; WC, waist circumference.

**Table 1 jpm-09-00013-t001:** Participants’ characteristics according to lower and higher circulating n-6/n-3 ratio (*n* = 898).

	Total	Lower n-6/n-3	Higher n-6/n-3	*P*
Sample size, *n* (men/women)	898 (260/638)	449 (122/327)	449 (138/311)	0.24
Age (years)	22.7 ± 0.1	22.7 ± 0.1	22.7 ± 0.1	0.93
Smokers, *n* (%)	1 (0.1)	0 (0)	1 (0.2)	1.0
Physical activity level (MET)	7.5 ± 0.1	7.4 ± 0.1	7.7 ± 0.1	0.07
BMI (kg/m^2^)	22.7 ± 0.1	22.7 ± 0.2	22.7 ± 0.2	0.95
WC (cm)	73.7 ± 0.3	73.6 ± 0.4	73.9 ± 0.4	0.65
Racial/ethnic groups, *n* (%)				<0.0001
Caucasians	455 (51)	195 (44)	260 (58)	
East Asians	338 (37)	203 (45)	135 (30)	
South Asians	105 (12)	51 (11)	54 (12)	
n-6 concentrations (µg/mL) ^1^	800 ± 253	798 ± 257	807 ± 255	0.22
n-3 concentrations (µg/mL) ^1^	62.6 ± 35.1	77.3 ± 34.5	51.6 ± 19.5	<0.0001
Circulating n-6/n-3 ratio	13.0 ± 4.7	10.7 ± 2.4	15.4 ± 3.2	<0.0001
Caloric intake (kcal)	1947 ± 21	1936 ± 30	1958 ± 30	0.61
Dietary n-6 intake (g) ^1^	10.7 ± 6.3	10.4 ± 6.3	10.7 ± 6.4	0.04
Dietary n-3 intake (g) ^1^	1.21 ± 0.82	1.28 ± 0.91	1.15 ± 0.68	0.001
Dietary n-6/n-3 ratio ^1^	8.7 ± 3.0	8.0 ± 3.0	9.3 ± 0.4	<0.0001

Values are given as mean ± SE or percentage as appropriate. *P*-values for difference between circulating lower and higher n-6/n-3 ratio (based on the median) were estimated with a Student’s *t*-test (continuous variables) and chi-square or Fisher’s exact test (categorical variables). ^1^ For circulating and dietary n-6 and n-3 fatty acids and their ratio (non-normal distribution), values are given as median and interquartile range and differences between groups were estimated with a Wilcoxon rank sum test. Abbreviations: BMI, body mass index; MET, metabolic equivalent tasks; WC, waist circumference.

**Table 2 jpm-09-00013-t002:** Interactions between *NFK1B* rs11722146, rs1609798, and rs230511genotypes and racial/ethnic groups on anthropometric traits (*n* = 898).

	BMI (kg/m^2^)			WC (cm)				
Genotypes (*n*)	Caucasians (*n* = 455)	East Asians (*n* = 338)	South Asians (*n* = 105)	*P_int._*	Caucasians (*n* = 455)	East Asians (*n* = 338)	South Asians (*n* = 105)	*P_int._*
rs11722146				0.02				0.04
GG (392)	23.3 ± 0.3	21.8 ± 0.3	24.2 ± 0.5		77.0 ± 0.6	72.2 ± 0.6	77.8 ± 1.2	
GA (394)	23.5 ± 0.3	22.3 ± 0.2	23.8 ± 0.6		77.5 ± 0.6	73.0 ± 0.5	77.3 ± 1.6	
AA (112)	23.6 ± 0.5	22.3 ± 0.4	20.6 ± 1.2		78.0 ± 1.2	73.6 ±0.8	69.8 ± 3.1	
*P*	0.75	0.26	0.02		0.61	0.33	0.06	
rs1609798				0.003				0.02
CC (402)	23.4 ± 0.3	21.9 ± 0.2	24.5 ± 0.5		77.2 ± 0.6	72.3 ± 0.6	78.2 ± 1.3	
CT (377)	23.4 ± 0.3	22.3 ± 0.2	23.5 ± 0.6		77.2 ± 0.6	73.0 ± 0.5	76.6 ± 1.6	
TT (119)	23.7 ± 0.5	22.2 ± 0.4	21.0 ± 1.0		78.3 ± 1.1	73.4 ±0.8	71.7 ± 2.7	
*P*	0.89	0.44	0.01		0.66	0.42	0.09	
rs230511				0.01				0.03
GG (379)	23.3 ± 0.3	21.8 ± 0.3	24.4 ± 0.5		76.9 ± 0.6	72.4 ± 0.6	78.1 ± 1.2	
GA (396)	23.6 ± 0.3	22.2 ± 0.2	23.5 ± 0.6		77.6 ± 0.6	72.9 ± 0.5	76.7 ± 1.6	
AA (123)	23.5 ± 0.5	22.2 ± 0.3	20.6 ± 1.3		77.8 ± 1.1	73.4 ±0.8	69.6 ± 3.3	
*P*	0.67	0.40	0.02		0.64	0.54	0.06	

*P* for interactions (*P_int_*) and *P* for differences in BMI or WC between genotypes were tested using a general linear model adjusted for caloric intake, physical activity; BMI, body mass index; WC, waist circumference.

**Table 3 jpm-09-00013-t003:** Associations (*p* values) between the *NFKB1* genotypes, circulating lower/higher n-6/n-3 ratio groups, and their interaction on anthropometric traits (*n* = 898).

	MAF	BMI (kg/m^2^)			WC (cm)		
		Genotype	n-6/n-3	Interaction	Genotype	n-6/n-3	Interaction
rs11722146	0.34	0.60	0.42	0.82	0.33	0.38	0.72
rs13117745	0.12	0.69	0.12	0.41	0.73	0.14	0.80
rs1609798	0.25	0.79	0.35	0.99	0.74	0.21	0.84
rs4648090 ^1^	0.07	0.31	0.48	0.02	0.23	0.57	0.03
rs4648022 ^1^	0.05	0.11	0.43	0.06	0.12	0.43	0.04
rs1599961	0.38	0.34	0.11	0.35	0.19	0.06	0.27
rs230511	0.36	0.78	0.27	0.81	0.51	0.24	0.92
rs7674640	0.49	0.63	0.91	0.44	0.53	0.68	0.55
rs3774932	0.50	0.52	0.58	0.10	0.18	0.62	0.12

*P* values were estimated using a general linear model adjusted for caloric intake, physical activity, age, sex, and racial/ethnic groups. ^1^ Since the minor allele frequency was 7% and 5% for rs4648090 and rs4648022, respectively, and ≤5 individuals were in the recessive group for either the lower or higher n-6/n-3 ratio groups, a dominant model was used for both SNPs. Abbreviations: BMI, body mass index; MAF, minor allele frequency; WC, waist circumference.

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
