# Peer review of "The Association between Plasma Omega-6/Omega-3 Ratio and Anthropometric Traits Differs by Racial/Ethnic Groups and NFKB1 Genotypes in Healthy Young Adults"

_jpm, 2019, doi:10.3390/jpm9010013_

Reviewer 1 Report

Reviewer’s comments:

This is a well written paper, using a large sample of young adults from various racial/ethnic groups. Authors are evaluating the NF-kB gene variants in relation to PUFA concentrations and anthropometrics. Results suggest an interaction between circulating n-6/n-3 and two NFkB1 genotypes on both BMI and waist circumference. This topic fits well within the scope of the journal.

General comments:

In the Introduction, the authors make a case for linking PUFA intake and circulating concentrations with obesity. However, the role of NF-kB genotypes on obesity is conceptually less clear, whereas relation with inflammation would be more direct. It is unclear what the authors’ hypothesis is.

The other concern is in regard to the study design to address the primary aim stated by the authors. On average, BMI was 22.7 kg/m2 and WC was 73.7 (although this includes both men and women), with limited distribution. This really hinders the full assessment of these associations. Importantly, no associations between n-6/n-3 and BMI or WC were observed. Similarly, genotypes were not associated with differences in BMI or WC. These findings weaken enthusiasm to some extent.

Specific comments:

Methods – Line 109 – T1D was exclusionary. Please comment on other known metabolic or immunological conditions that may relate here (T2D, cancer, etc.).

Results:

Section 3.2 –

There are several comparisons of BMI across racial/ethnic groups. It is important to point out that for a given BMI, Asians have a higher %body fat, and thus different BMI criteria for weight classification (WHO, Lancet 2004; Wang J, AJCN 1994). This may impact interpretation and warrants mentioning.

Authors note that NFkB1 genotype frequency differed by racial/ethnic groups. These frequencies seems to be important information worth informing.

Author Response

Comment 1: In the Introduction, the authors make a case for linking PUFA intake and circulating concentrations with obesity. However, the role of NF-kB genotypes on obesity is conceptually less clear, whereas relation with inflammation would be more direct. It is unclear what the authors’ hypothesis is.

 Response: We agree with the reviewer that the premise and hypothesis of the study could be clarified. We have added the following information (underlined below):

Lines 73 to 79: “Transgenic Fat-1 mice were also shown to have reduced inflammation through a marked attenuation of the nuclear factor NF-kB activity [11], a transcription factor acting as an upstream regulator of the inflammatory response [13]. Lines of evidence suggest that one of the proposed mechanisms linking n-6 and n-3 PUFA to obesity and obesity-related complications is inflammation [14]. Since NF-kB can be modulated by n-6 and n-3 fatty acids [15–20], variations in this gene may potentiate or attenuate the n-6/n-3 mediated inflammatory response and its subsequent effects on obesity-related mechanisms.

Lines 88-91: “Thus, we hypothesize that inconsistent results in human observational studies examining the relationship between PUFA intake and obesity-related phenotypes may partly be explained by genetic variations in NF-kB given its upstream role in inflammation and its known regulation by PUFA.”

Comment 2: The other concern is in regard to the study design to address the primary aim stated by the authors. On average, BMI was 22.7 kg/m2 and WC was 73.7 (although this includes both men and women), with limited distribution. This really hinders the full assessment of these associations. Importantly, no associations between n-6/n-3 and BMI or WC were observed. Similarly, genotypes were not associated with differences in BMI or WC. These findings weaken enthusiasm to some extent.

 Response: We acknowledge this is a limitation and have added the following sentence in the discussion (lines 389 to 392): “We did not find an association between n-6/n-3 ratio and obesity-related anthropometric traits. However, our ability to assess this association may have been hampered by the limited distribution of BMI and WC in our population of healthy young individuals.”

Comment 3: Methods – Line 109 – T1D was exclusionary. Please comment on other known metabolic or immunological conditions that may relate here (T2D, cancer, etc.).

 Response: We have expanded the list of examples of exclusion criteria applied as follows (lines 114 to 117): “Participants were excluded from the study if they were pregnant or breastfeeding due to hormonal and physiological adaptations occurring during these periods [24]; if they had chronic diseases that could influence inflammation (e.g., type 1 or 2 diabetes, cancer, cardiovascular disease); or if they could not provide a venous blood sample.”

Comment 4: Results: section 3.2 – There are several comparisons of BMI across racial/ethnic groups. It is important to point out that for a given BMI, Asians have a higher %body fat, and thus different BMI criteria for weight classification (WHO, Lancet 2004; Wang J, AJCN 1994). This may impact interpretation and warrants mentioning.

 Response: One sentence addressed differences in regional mass and body composition between races/ethnicities (lines 347 to 350), and as suggested, we have added the following sentence referencing the WHO expert guidelines (2004) (lines 350-351): “Indeed, lower cut-off points for higher-risk categories of BMI and WC have been proposed for Asian populations [31].”

Comment 5: Authors note that NFkB1 genotype frequency differed by racial/ethnic groups. These frequencies seems to be important information worth informing.

Response: A new supplemental Table 2 provides the frequencies by NFKB1 variants in the different racial/ethnic groups.

Reviewer 2 Report

I enjoyed reading this comprehensive study on the interaction between diet, plasma fatty acids, weight and genotype. Overall a very good study.

The authors discuss the relationship between NFKB and its signaling of the inflammatory response thereby potentially contributing to obesity. The premise of the research suggests particular NFKB genotypes will influence obesity. I am less clear from the manuscript how the plasma and dietary intake of omega 3 and 6 fits in? Do the authors hypothesize that a high omega 3 diet offsets the genotypic differences? Or vice versa? The notion has been touched on in the discussion, but the argument needs to be developed further and introduced in the introduction.

The results appear to be mentioned in table and text form on multiple occasions. Numbers should be mentioned once only, either chose the table or text format. Don't double up the results. Refer to the table if necessary.

Figure one and two need to be better quality inserts. Can both figures be recreated in GraphPad or another suitable graphing software? Remove the word "Panel", simply refer to A and B.

Table 3- remove the internal x-axis lines (keep header), as it is presented for table 1 and 2.

Author Response

Comment 1: The authors discuss the relationship between NFKB and its signaling of the inflammatory response thereby potentially contributing to obesity. The premise of the research suggests particular NFKB genotypes will influence obesity. I am less clear from the manuscript how the plasma and dietary intake of omega 3 and 6 fits in? Do the authors hypothesize that a high omega 3 diet offsets the genotypic differences? Or vice versa? The notion has been touched on in the discussion, but the argument needs to be developed further and introduced in the introduction.

Response: Reviewer #1 made a similar comment, which has been addressed above. Please see our response to Reviewer #1, Comment 1.

Comment 2: The results appear to be mentioned in table and text form on multiple occasions. Numbers should be mentioned once only, either chose the table or text format. Don't double up the results. Refer to the table if necessary.

Response: We have removed duplicated numbers in the text of the results section and now refer to tables.

Comment 3: Figure one and two need to be better quality inserts. Can both figures be recreated in GraphPad or another suitable graphing software? Remove the word "Panel", simply refer to A and B.

Response: Higher quality figures have been inserted and the word “panel” was removed.

Comment 4: Table 3- remove the internal x-axis lines (keep header), as it is presented for table 1 and 2.

Response: The line was removed in the header of table 3.